# Effect of freeze-thaw cycles on mechanical performance of loess soil stabilized with nano magnesium oxide

Peng Hu[1,2], Shufeng Chen [id]³*, Zhao Duan[1], Nian-qin Wang[1], Ye Hao[2], Xian Wang[2]

**1** College of Geology and Environment, Xi'an University of Science and Technology, Xi'an, China, **2** Shaanxi Tiandi Geology Co., Ltd, Xi'an, China, **3** Shaanxi Key Laboratory of Safety and Durability of Concrete Structures, Xijing University, Xi'an, China

* csf1127@qq.com

## Abstract

Construction in northwest China is generally packed with issues linked to loess soil with poor engineering properties and day-night and seasonal freeze-thaw (FT) actions. This study explored the potential benefits of nano-MgO (NM) as an innovative solution for improving mechanical properties of loess. To this end, a series of unconfined compression test (UCT) and nuclear magnetic resonance tests (NMRT) were conducted. Results showed that the unconfined compressive strength (UCS) exhibited an a "rise-fall" trend with the addition of NM. An optimum dosage of 2% NM is expected to bring about 71.9% and 143.5% strength gain for non-FT and FT samples, respectively. Meanwhile, the FT-induced strength reduction ratio decreased from 56.3% to 38.1% with NM content from 0 to 2%. These illustrated that NM can be very effective in improving mechanical performance and alleviating freeze-thaw damage. On the other hand, deformation modulus presented similar trends with UCS, while failure strain behaved in a reverse way. Accordingly, empirical models for UCS, as well as its relationships with modulus and failure strain, were established and validated by literature data. Furthermore, nuclear magnetic resonance tests revealed that adding NM could increase the proportion of bound water with intensive interaction, yielding improved performance and durability. This investigation shows that NM represents an alternative to cement for soil stabilization, and provides scientific support for the construction design in cold regions.

## 1. Introduction

Loess is silt-sized aeolian-deposit widely distributed from west- to east-China over the strip between 30 °N and 48 °N latitude [1,2]. Nature loess is featured with large porosity, weak and soluble inter-particle bonds, and high freeze-thaw susceptibility, which brings serious subsidence, settlement, and cracks to the structures built on it [3,4]. This, in turn, drove the rise of stabilization techniques for boosting the

**Data availability statement:** All relevant data are within the manuscript and its Supporting Information files.

**Funding:** This research is supported by the National Natural Science Foundation of China (No. 12102367 to SC); the Special Fund for the Launch of Scientific Research in Xijing University (No. XJ23T01 to SC); the Horizontal Project of Xijing University and Shaanxi Coal Geology Group Co., Ltd. (No. 2023610002003171 to SC, SMDZ-2023CX-15 to PH). The funders had no role in study design, data collection and analysis, decision to publish, or preparation of the manuscript.

**Competing interests:** The authors have declared that no competing interests exist.

engineering performance of loess. Worldwide, conventional cementing additives (cement, lime, etc.) have garnered wide extent of use in a number of disciplines [5,6]. Several studies are specifically concern about loess or loess-like soil and achieved satisfactory improvements [7–9]. However, overuse of these materials causes rising environmental concerns due to their energy-intensive production process [10,11]. In the context of China's 2060 carbon neutrality pledge, low-carbon-footprints additive has become increasingly sought by Chinese industries.

In recent years, the development of novel soil stabilization techniques has opened up eco-friendly alternatives for construction engineers and designers, such as industrial by-products [12], agriculture wastes [13], microbial induced calcite precipitation [14], and nanomaterials [15]. The nanomaterial stabilization techniques are a matter of great concern due to their high performance resulting from the high specific surface areas, unsaturated bonds, and surface activity of nanoparticles; meanwhile, more and more attentions are given to nano-MgO (NM) as a green and low-carbon stabilizers [16,17]. Many scholars have been devoted to substitute or partially substitute NM for traditional stabilizers. It was revealed that a small amount of NM (less than 5%) could bring significant mechanical improvements owing to the cementation, pore filling, and water absorption effects of NM particles [18–26]. However, the application of NM in soil stabilization is still in its early stages, with some controversial aspects remained. For instance, some researchers reported adverse effects of NM on soil strength in high level of nanomaterial content [18,19,21,27,28]. Therefore, further research is needed in this emerging field, especially in its performance for modifying loess soil.

Severe fluctuation of atmospheric temperatures is a common occurrence in loess territories. Periodic freeze-thaw (FT) to natural loess would cause particle aggregation, changes pore size distribution, break inter-particle bondage and bring degeneration of soil geotechnical properties (e.g., hydraulic conductivity, strength, stiffness) [29,30], even for the stabilized loess by various additives [7,31]. As for nano-stabilized soils, the impacts of FT cycles should not be ignored in view of the improving effects of nanoparticles with strong FT susceptibility. Kalhor et al. [27] found 63% UCS gain of a clayey soil after treated with 2% nano-$SiO_2$; besides, the FT-induced strength reduction reduced from 64% to 42%. Kakroudi et al. [32] demonstrated that the silty sand treated with an optimal dosage of 10% nano-$SiO_2$ and 1% basalt fiber reveals considerable improvements in the static and dynamic performance, as well as in its freeze-thaw resistance. Chen et al. [33] studied the effects of nano-MgO in the dynamic properties of loess soil subjected to FT cycles, and conclude that reasonable dosage (2.5%) can improve the dynamic modulus and alleviate the detrimental effects of FT cycles. In sum, the study on the performance of nanomaterials modified soil in FT circumstance is still very limited, and existing studies in this field tend to be focus on nano-$SiO_2$ or dynamic properties. The knowledge on the mechanical performance of nano-MgO modified soil under FT circumstance is still not adequate, and requires further exploration.

In light of the above, the present work aims at exploring the performance of NM as a green and low-carbon binder for enhancing the mechanical properties of loess

subjected to FT cycles. The unconfined compression test (UCT) was carried out to investigated the strength and deformation characteristics of NM-treated loess. The strength and stiffness characteristics of the modified loess, as well as its freeze-thaw resistance, were investigated and quantitatively evaluated. Furthermore, nuclear magnetic resonance test (NMRT) was invoked to reveal the stabilizing mechanism of NM in terms of water-state variation inside the soil. The findings of this work will provide reference for the application of nanomaterials in subgrade engineering in cold regions.

## 2. Materials and methodology

### 2.1. Materials and mixtures

Loess soil adopted in this work was available from a construction site in Chang'an distinct, Xi'an, China. It is a mustard yellow, loose, uniform, late Pleistocene ($Q_3$) loess with basic properties listed in Table 1. The utilized NM was produced by Messi Biotechnology Co. Ltd., China. It was a white, nano-scale, high purity powder with spherical particle size of 30–50 nm and specific surface area of 15–30 m²/g (Fig 1a). Several key features of the nano-MgO were listed in Table 2.

Standard proctor compaction tests (PCT) were conducted to reveal the compression characteristics of NM-treated loess (NML). The result in Fig 2 indicates that the addition of NM up to 4% contributes to an increase in optimum moisture content $w_{opt}$ from 19.8% to 23.9%, while a decrease in maximum dry density $\rho_{dmax}$ from 1.64 g/cm³ to 1.57 g/cm³. The rise of $w_{op}$ is due to the high water-absorption ability of NM. Also, the enhanced interparticle bond by NM prevented easy compaction which results in reduction in $\rho_{dmax}$. Similar trends were also reported in the studies on other nano-scale additives [25,27,34].

### 2.2. Experimental program and sample preparation

Unconfined compaction test (UCT) was performed to investigate the mechanical characteristics of NML concerning FT cycles, NM content, and water content. Several NMR tests were conducted for uncovering the stabilizing mechanism. The mixing design and corresponding test items are detailed in Table 3.

Before sampling, raw loess was firstly air-dried, grinded, sieved (2.0 mm), and then oven-dried for 24 hours. Based on the mix design, the materials were combined together and further manually stirred sufficiently until uniformity was ensured. In either case, distilled water was added according to the optimum moisture content from PCT for achieving maximum dry density. The mixture was humectation for 24 hours prior to the next compaction. Thereafter, the materials were packed into a cylindrical mold (39.1 mm diameter and 80 mm height) in three layers (Fig 1b). For UCT, two samples

**Table 1. Basic properties of the loess soil.**

| Properties | Index | Unit | Value |
|---|---|---|---|
| Physical index | Specific gravity | – | 2.64 |
| | Plastic limit | % | 20.5 |
| | Liquid limit | % | 34.2 |
| | Plastic index | – | 13.7 |
| Grain size distribution | >2.0 mm (Gravel) | % | 0.4 |
| | 2~0.05 mm (Sand) | % | 9.7 |
| | 0.05~0.002 mm (Silt) | % | 66.1 |
| | <0.002 mm (Clay) | % | 23.8 |
| Mineral components | Quartz | % | 41 |
| | Feldspar | % | 29 |
| | Muscovite | % | 14 |
| | Montmorillonite | % | 9 |
| | Illite | % | 7 |

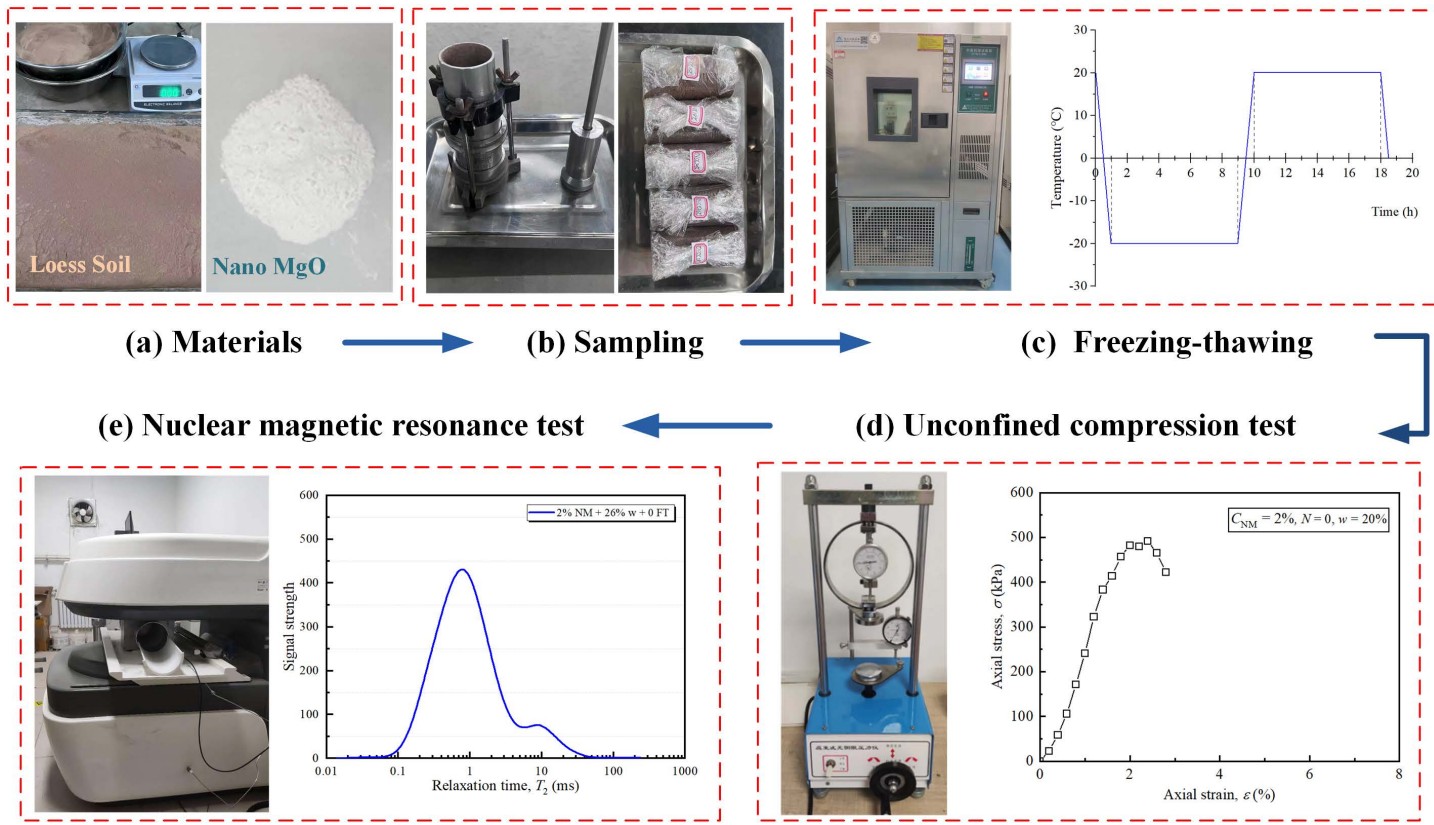

**Fig 1. Schematic diagram of laboratory test (a) Materials, (b) Sampling, (c) Freeze-thaw chamber, (d) Unconfined compression test, (e) Nuclear magnetic resonance test.**

**Table 2. Features of the used MgO.**

| Properties | Unit | Value |
|---|---|---|
| Particle shape | – | Spherical |
| Purity | % | 99.9 |
| Melting point | °C | 2850 |
| Boiling point | °C | 3600 |
| Density | g/cm³ | 3.58 |
| Formula weight | g/mol | 40.30 |
| Particle size | nm | 30-50 |
| Specific surface area | m²/g | 15-30 |
| Bulk density | g/cm³ | 0.74 |

were fabricated at each test condition to ensure consistency. To minimize the discreteness of test results, the difference between the actual and target dry density was controlled within 0.01 g/cm³, and the difference in water content was within 0.1%.

Subsequently, the samples were isolated with three plastic layers and placed in the curing chamber for 7 days. After curing, the samples were exposed to FT cycles in a programmable test chamber with temperature range of -40–100°C

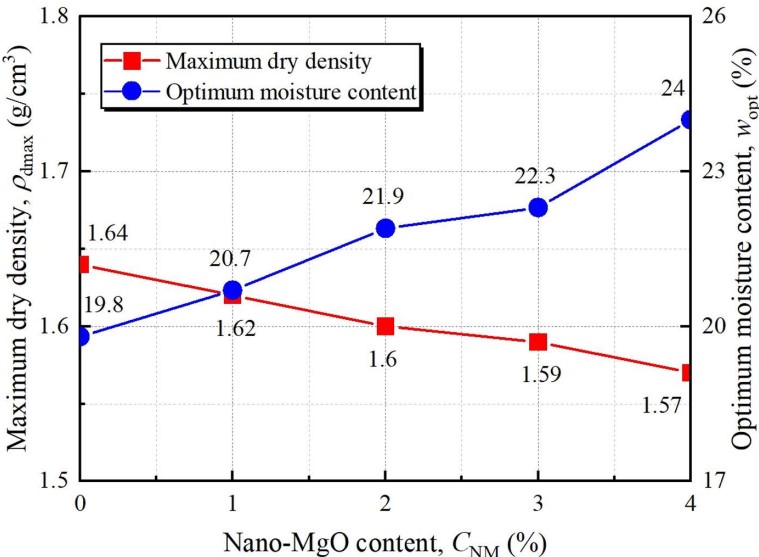

**Fig 2. Results of Standard proctor compaction test.**

**Table 3. Testing programs.**

| Test items | Series | Initial conditions | | |
| --- | --- | --- | --- | --- |
| | | NM content (%) | Water content (%) | Freeze-thaw cycles (-) |
| UCT | S1 | 0 | 20 | 0, 1, 3, 6, 10 |
| | S2 | 1 | | |
| | S3 | 2 | | |
| | S4 | 3 | | |
| | S5 | 4 | | |
| | S6 | 2 | 14 | 0, 1, 3, 6, 10 |
| | S7 | | 17 | |
| | S8 | | 20 | |
| | S9 | | 23 | |
| | S10 | | 26 | |
| NMRT | S11 | 0 | 20 | 0, 10 |
| | S12 | 2 | | |
| | S13 | 2 | 20 | 0, 10 |
| | S14 | | 26 | |

and ± 0.5°C in precision (Fig 1c). According to the local winter temperature, freezing and thawing process in this study were respectively conditioned at -20°C and 20 °C, and lasted for 9 hours for completely frozen or thawed.

## 2.3. Instruments

The employed unconfined compression apparatus was the strain-controlled YYW-2 model manufactured by Nanjing Soil Instrument Company, Jiangsu, China (Fig 1d). Following the standard guidelines of TMSHE JTG E-2007, UCTs were carried out under constant axial displacement rate of 0.8 mm/min, and continued until sample failure or 10% of axial strain

achieved. Afterwards, the key mechanical parameters, including UCS, deformation modulus and failure strain, can be obtained. For the scenarios listed in Table 3, S1-5 are to reveal the effect of NM and FT cycles with constant water content, while S6-10 are to evaluate the effect of water content and FT cycles with constant NM content.

The NMRTs were performed using a Low-field NMR system supplied by Niumag Instrument Corporation, Suzhou, China (Fig 1e). The room temperature was controlled at 32 ± 0.5 °C for testing accuracy. The NMR apparatus possesses magnetic field strength of 0.3 T permanent and resonance frequency of 1~30 MHz, with an accuracy of 0.1 Hz. The repetition time and echo intervals were 1000 ms and 0.30 ms, respectively. The number of echoes was 8000, and the radio of frequency diameter was 60 mm. The sample series S11-14 were contained in a polytetrafluoroethylene mold with 52 mm in diameter and 100 mm in height, and then placed in the testing tube to collect the signal and obtain a $T_2$ curve. The relaxation signals were collected by NMR Analysis software (Suzhou Niumag Analytical Instrument Co., Ltd.).

## 3. Results and discussion

### 3.1. Unconfined compressive strength of NML

Fig 3a illustrates the unconfined compressive strength (UCS) of NML with varying NM dosages and FT cycles. It is observed that the UCS increased with increasing NM dosage at first, and then decreased. That grants the maximum UCS of 519.6 kPa, 440.7 kPa, 378.4 kPa, 347.7 kPa, 323.3 kPa for NML samples exposed to 0, 1, 3, 6, 10 times of FT cycles, respectively. Almost all of the peak strength were achieved at NM content of 2%, which was ascertained as the optimum dosage for loess stabilization. Through literature review, similar "rise-fall" pattern was also observed in soil stabilization researches on other nanomaterials [20,25,28,34].

The NM-induced "rise" in UCS can be attributed to the cementation effects of NM. Owing to large specific surface area, NM has more surface atomic number with mass dangling bond, which combined with soil particles and exerted considerable interparticle cementing effects [29]. Nevertheless, for a given water content, excessive addition of NM not only makes the mixture less compressible, but also caused over consumption of water which reduce the interparticle bondage of the mixture. These effects ultimately negated the UCS growth by the cementing effect of NM, resulting in reduction in UCS. Besides, overuse of nano-materials is also prone to cause unstable lumps and weak area resulting from uniformly dispersed NM, which would lead to strength loss, either. As evident from Fig 3a, the effects of FT on UCS varied depending

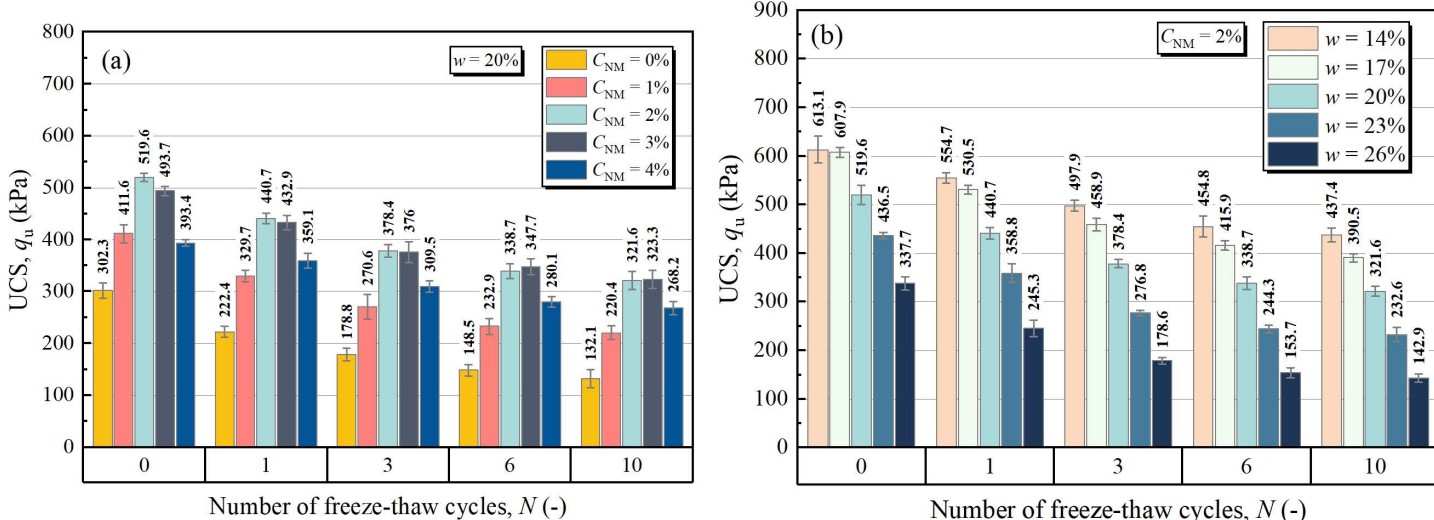

**Fig 3. Unconfined compressive strength for different: (a) NM contents and FT cycles, (b) water contents and FT cycles.**

on NM percentage. With FT cycles from 0 to 10, the UCS for NM content of 0%, 1%, 2%, 3%, 4% respectively reduced by 56.3%, 46.4%, 38.1%, 34.5%, 31.8%. On the other aspect, as NM from 0% to 2%, NML samples subjected to 0, 1, 3, 6, 10 FT cycles exhibited UCS gains of 71.9%, 98.1%, 111.6%, 128.1%, 143.5%, respectively. Suffice it to say that the improving effects of NM lies not only in the strength of loess, but also in the resistance to FT cycles.

Fig 3b demonstrates the UCS for NML with varying water contents and FT cycles. It is clearly that both water content and FT cycles exerted negative effects on UCS. As water content increased from 14% to 26%, the UCS for 0, 1, 3, 6, 10 FT cycles reduced by 44.9%, 55.8%, 64.1%, 66.2%, 67.3%, respectively. In terms of FT cycles, the UCS values for water content of 14%, 17%, 20%, 23%, 26% respectively dropped 28.7%, 35.8%, 38.1%, 46.7%, 57.7% after 10 times of freezing and thawing. The phenomena above signal the presence of mutually promotive interaction between adding water and FT cycles [35]. Actually, the detrimental effects of FT on soil strength are related to the phase change of water into ice under subzero temperatures, which brings pores expansion and crystallizing pressure upon the particles around. Apparently, the higher the water content, the more evident the impact of FT cycles. In other words, the effects of FT cycles can be alleviated in practice by preventing water intrusion.

## 3.2. Failure strain of NML

Fig 4a demonstrates the failure strain of samples with different FT cycles and NM contents. It is seen that the failure strain decreased as NM increased from 0% to 2%, and continued to increase up to a NM content of 4%. If the cases of 3 FT cycles are considered, the failure strain firstly decreased from 4.6% to 2.6%, and then increased to 3.4% with increasing NM content. In all cases, as FT cycle increased, the failure strain presented an increasing trend. Furthermore, this FT-induced increase in failure strain tended to be less pronounced with increasing NM content. For example, as FT cycles from 0 to 10, the failure strain of natural loess increased by 83.3% (i.e., from 3.6% to 6.6%), while that of 4% NM treated samples were just 48.1% (i.e., from 2.7% to 4.0%). That implied that NM could help weaken the impact of cyclic freezing and thawing. Fig 4b illustrated the effects of FT cycles and water content on failure strain of samples at a fixed NM content of 2%. Generally, the failure strain presented an increasing trend with FT cycles and water content. The FT-induced increment of failure strain is more pronounced at higher water content. Specifically, the average $\varepsilon_f$ value for 26% water content is 2.16 times higher than corresponding value for 14% water content.

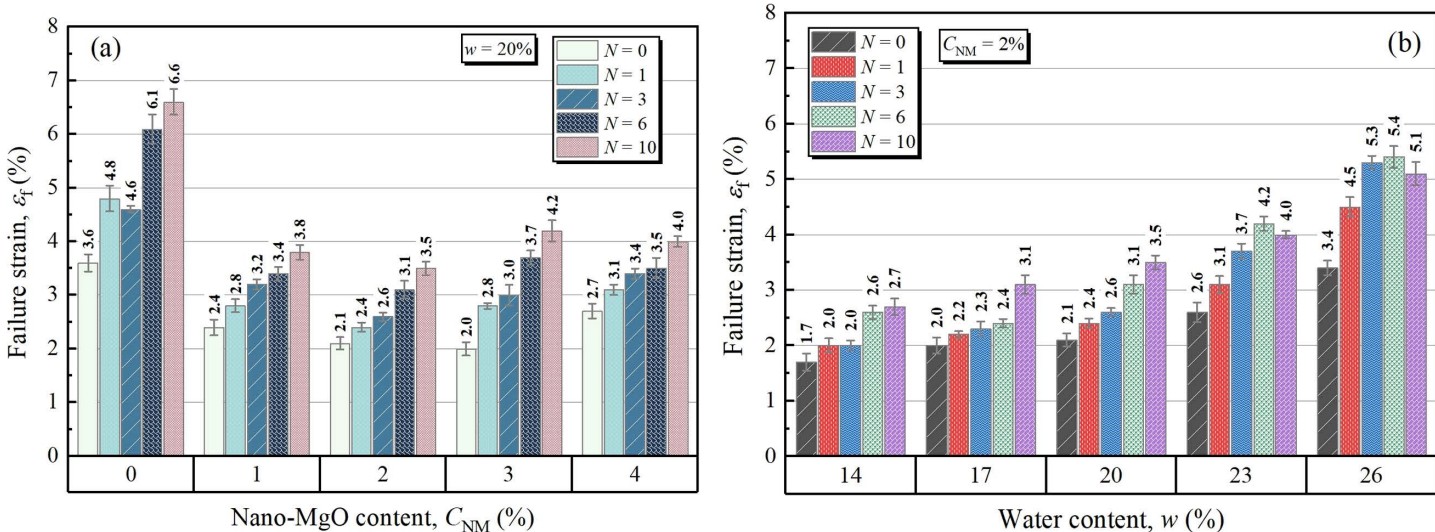

**Fig 4. Failure strain for different: (a) NM contents and FT cycles, (b) water contents and FT cycles.**

Fig 5 exhibits the relation between failure strain and UCS under varying FT conditions. As seen from the figure, the failure strain of NML presents an exponential decreasing trend with the increase of UCS. With regression analysis, this relationship can be expressed by natural logarithmic function,

$$\varepsilon_f = \kappa \cdot \ln q_u + \zeta \tag{1}$$

where, $\varepsilon_f$ and $q_u$ is expressed in % and MPa, respectively. $\kappa$ and $\zeta$ are coefficients listed in Fig 5. The fair determination coefficient ($R^2 = 0.905$) not only indicates the good fitting results of Eq (1), but also suggests that the $\varepsilon_f$ - $q_u$ relation is independent of NM content, water content, and FT cycles to some extent. This characteristic brings convenience to the application in engineering practice. Wang et al. [17] and Kalhor et al. [27] investigated the stabilization effects of reactive MgO (0–9%) and nano-SiO$_2$ (0–3%) on soils by performing UCS tests. Valuable experimental data extracted from these studies were also fitted to Eq (1) with the model coefficients listed in Fig 5. In most cases, the fitting attains high determination coefficient, $R^2$, which validates the applicability of the model.

### 3.3. Deformation modulus of NML

Modulus is an important parameter that describes their deform behavior under stress. In view of the nonlinear stiffness behavior of soil, the deformation modulus, $E_{50}$, was investigated in present work. $E_{50}$ represents the stiffness corresponding to 50% of failure strain, and can be determined from UCS test by the following equation:

$$E_{50} = \frac{2\sigma_{1/2}}{\varepsilon_f} \tag{2}$$

where $\varepsilon_f$ is the failure strain, $\sigma_{1/2}$ is the stress corresponding to the half of failure strain.

Fig 6a depicts the effects of NM content and FT cycles on the deformation modulus of NML at a fixed water content of 20%. With cyclic freezing and thawing, significant drop in $E_{50}$ was observed. After 10 times of freezing-thawing, the $E_{50}$ of sample with 2% NM content took a 71% reduction (i.e., from 33.0 MPa to 12.6 MPa). While, as the NM content increased, the $E_{50}$ presented a "rise and fall" trend, reaching its maximum around 2% of NM content. Fig 6b illustrates the effects of water content and FT cycles on the $E_{50}$ of samples at a fixed NM content of 2%. As is shown, the $E_{50}$ decreased

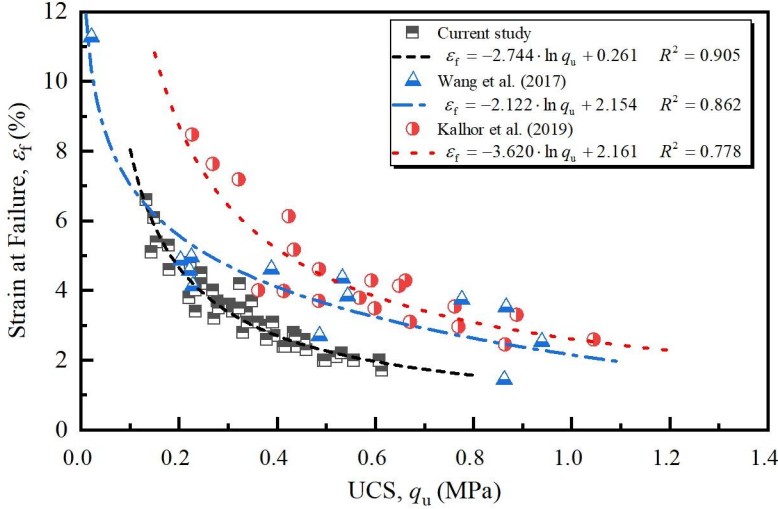

**Fig 5. Relationship between failure strain s and UCS.**

significantly with an increase in the number of FT cycles. This effect becomes more significant with the addition of water content in samples. For example. At $w$=26%, the $E_{50}$ of unfreezing-thawing sample took an 71% reduction (i.e., 13.5 MPa to 3.9 MPa), which is higher than corresponding reduction rate of 55% for $w$=14%. It indicated that the FT damage is enhanced with increasing water content.

Fig 7 presents the deformation modulus versus UCS for different FT cycles. It is observed that the $E_{50}$ of NML exhibits an increasing trend with the increase of UCS, and the increasing growth rate was also noted. Accordingly, a quadratic regression equation passing through the origin was used to describe this relationship:

$$E_{50} = \eta \cdot q_u^2 + \beta \cdot q_u \tag{3}$$

where, $E_{50}$ and $q_u$ are expressed in MPa. Regression analysis of the experimental data in current study and literature yielded coefficients $\eta$, $\beta$ and $R^2$ presented in Fig 7. The determination coefficients of 0.974, 0.866 and 0.950 for the data from current study and literature suggested the goodness of fitting as well as the independency of $E_{50}$ - $q_u$ relation to nanomaterial content and FT cycles.

## 3.4. Empirical expression for UCS

As previously discussed, cyclic freezing-thawing exerted negative effect on the mechanical behavior of NML, which was correlated with NM and water content. For describing these relationships, the durability index, $I_d$, was introduced as follows,

$$I_d = \frac{q_u}{q_{u-0}} \cdot 100\% \tag{4}$$

where $q_u$ is the UCS of NML under FT cycles; $q_{u-0}$ is the corresponding value of pre-freezing-thawing samples.

Fig 8a illustrates the variation of durability index for various NM and water contents. The detrimental effect of FT on strength is well represented by the declining trend of durability index. With cyclic FT, the durability index takes the maximum decreasing rate at the first cycle, and decreases with lower rates in the subsequent cycles. After about 6 times of

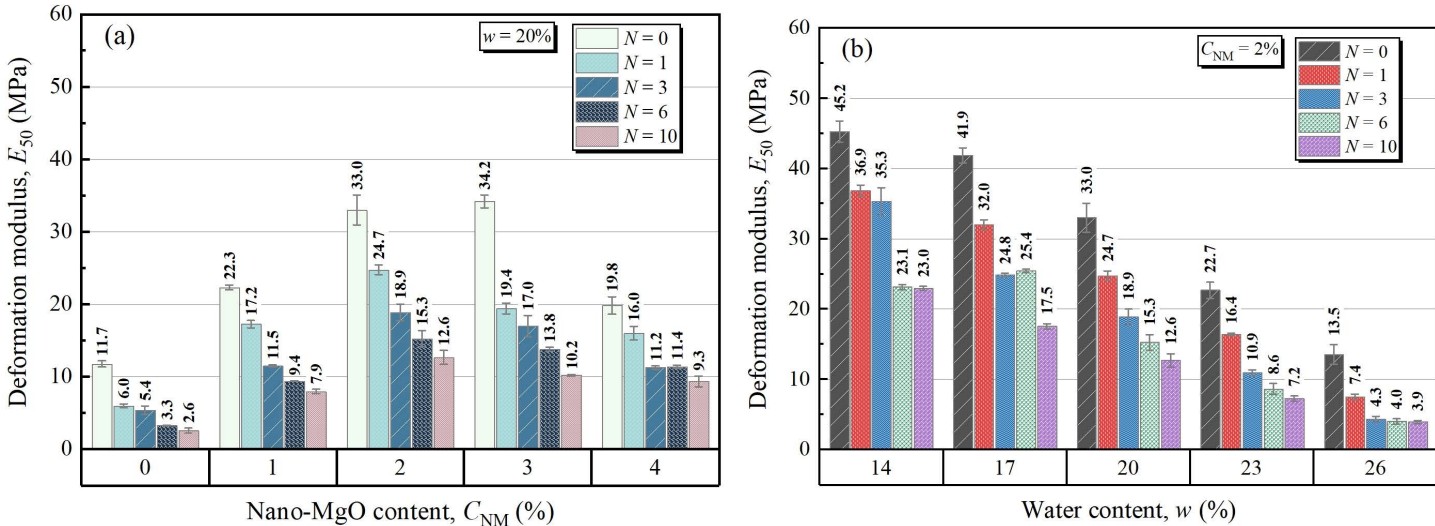

**Fig 6. Deformation modulus for different: (a) NM contents and FT cycles, (b) water contents and FT cycles.**

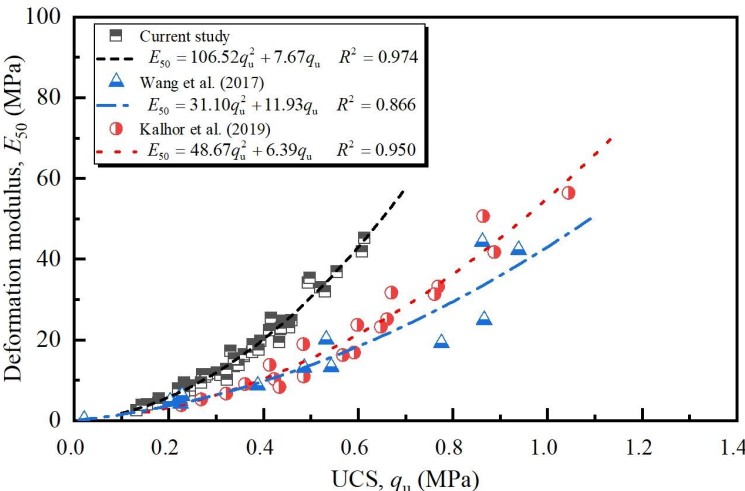

**Fig 7. Relationship between deformation modulus and UCS.**

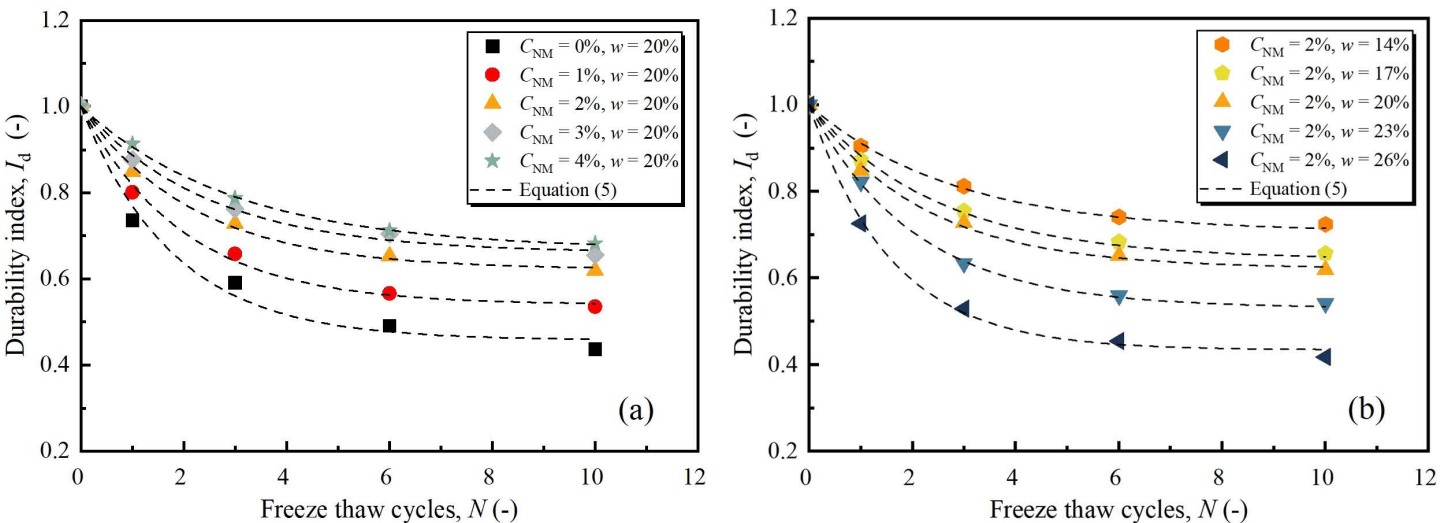

**Fig 8. Durability index for different: (a) NM contents and FT cycles, (b) water contents and FT cycles.**

FT cycles, the decrease of durability index became quite gentle. It suggests a balance was preliminarily achieved under FT circumstance. As for the effects of NM and water, visual observation in Fig 8b clearly states that FT-induced damage becomes more severe under higher water content levels, while turns gentle with the addition of NM. To be specific, the durability index from the maximum FT cycles increased from 43.7% to 68.2% with NM content from 0% to 2%, and decreased from 55.1% to 32.7% as water content from 14% to 26%.

From the results in Fig 8, the durability index can be expressed by a logarithmical relationship as follows:

$$I_d = 1 + a\left(e^{b \cdot N} - 1\right) \tag{5}$$

where, $a$ and $b$ are regression coefficients. The higher the value of $a$ (positive), the greater strength loss FT caused. While, a higher value of $b$ (negative) indicates a faster decreasing rate with FT cycles. 15 pairs of $I_d$ -$N$ relations collected from present work and relevant literature were fitted according to Eq (5), and the results were tabulated in Table 4. The high determination coefficients ($R^2$) of the analysis indicated the applicability of Eq (5) to describe the strength reduction under FT cycles. As in shown, it is notable that the $a$ of NML decreased with increasing NM content, while the corresponding $b$ value increased. By contrast, as the water content increased, the $a$ of NML increased, but the corresponding $b$ value decreased. This observation is in line with the results of Liu et al. [36]. The parameters $a$ and $b$, for the results of Kalhor et al. [27], first increased and then decreased with the addition of nano-$SiO_2$. This is mainly due to the rise of nano-$SiO_2$ content paralleled by an increase in water content.

Multiple regression analysis was applied for determining the UCS of NML. Quite clearly, based on Eqs (4) and (5), UCS can be expressed a function of $q_{u-0}$ and $N$ as follows,

$$q_u = q_{u-0} \cdot \left[ 1 + a\left( e^{b \cdot N} - 1 \right) \right] \tag{6}$$

The parameters $q_{u-0}$, $a$ and $b$ can be estimated from their correlation with $C_{NM}$, $w$ and $\bar{q}_{u-0}$, where $\bar{q}_{u-0}$ was standard UCS selected as the value of $q_{u-0}$ for $C_{NM} = 2\%$ and $w = 20\%$ (i.e., 519.6 kPa).

Fig 9a–9c separately illustrates the variations of $q_{u-0}/\bar{q}_{u-0}$, $a$ and $b$ against $C_{NM}$ and $w$. By simple linear or quadratic regression analysis, the empirical models for $q_{u-0}$, $a$ and $b$ are developed in terms of $C_{NM}$ or $w$. The fitness and significance of the developed equations are evaluated by coefficient of determination ($R^2$). The high $R^2$-values ranging from 0.913 to 0.998 indicates the strong correlations between these factors.

Taking the equations in Fig 9 into Eq (6), comprehensive functions considering three governing factors, $C_{NM}$, $w$, and $N$, are established as follows,

Model 1 is established for predicting the strength of NM-treated loess at a fixed $w$ of 20%, the optimum water content of loess soil,

**Table 4. Fitting results of the data from current study and literatures.**

| Soil | Additive | Dosage | Water content | FT cycles | $a$ | $b$ | $R^2$ | Reference |
|------|----------|--------|---------------|-----------|-----|-----|-------|-----------|
| Loess | Nano-MgO | 0% | 20% | 0~10 | 0.542 | -0.555 | 0.986 | Current work |
| | | 1% | | | 0.46 | -0.503 | 0.996 | |
| | | 2% | | | 0.378 | -0.455 | 0.996 | |
| | | 3% | | | 0.34 | -0.405 | 0.994 | |
| | | 4% | | | 0.332 | -0.332 | 0.999 | |
| | | 2% | 14% | | 0.293 | -0.358 | 0.999 | |
| | | | 17% | | 0.357 | -0.398 | 0.997 | |
| | | | 20% | | 0.378 | -0.455 | 0.996 | |
| | | | 23% | | 0.469 | -0.489 | 0.999 | |
| | | | 26% | | 0.566 | -0.628 | 0.998 | |
| Clay | Nano-$SiO_2$ | 0% | 17.6% | 0~9 | 0.778 | -0.215 | 0.897 | Kalhor et al., 2019 [27] |
| | | 1% | 19.5% | | 0.755 | -0.134 | 0.939 | |
| | | 2% | 21.9% | | 0.803 | -0.088 | 0.953 | |
| | | 3% | 24.1% | | 0.719 | -0.136 | 0.937 | |
| Clay | Cotton straw Fiber | 0% | 34.3% | 0~20 | 0.36 | -0.469 | 0.995 | Liu et al., 2020 [36] |
| | | 0.2% | | | 0.25 | -0.362 | 0.992 | |
| | | 0.4% | | | 0.23 | -0.322 | 0.973 | |

$$q_{u} = 519.6 \cdot \left[ 1 + \left( -5.401 \cdot C_{NM} + 0.518 \right) \left[ e^{\left( 5.440 \cdot C_{NM} - 0.559 \right) \cdot N} - 1 \right] \right]$$
$$\cdot \left[ -853.55 \cdot \left( C_{NM}^2 - 0.02^2 \right) + 39.23 \cdot \left( C_{NM} - 0.02 \right) + 1 \right] \tag{7}$$

According to the formula, the unconfined compressive strength of stabilized loess with different NM content and FT cycles can be calculated during construction design.

Model 2. is established for evaluate the freeze-thaw resistance of NM-treated loess at the optimum NM dosage, i.e., $C_{NM}$ of 2%,

$$q_{u} = 519.6 \cdot \left[ 1 + \left( 2.193 \cdot w - 0.026 \right) \left[ e^{\left( -2.103 \cdot w - 0.045 \right) \cdot N} - 1 \right] \right]$$
$$\cdot \left[ -21.90 \cdot \left( w^2 - 0.2^2 \right) + 4.13 \cdot \left( w - 0.2 \right) + 1 \right] \tag{8}$$

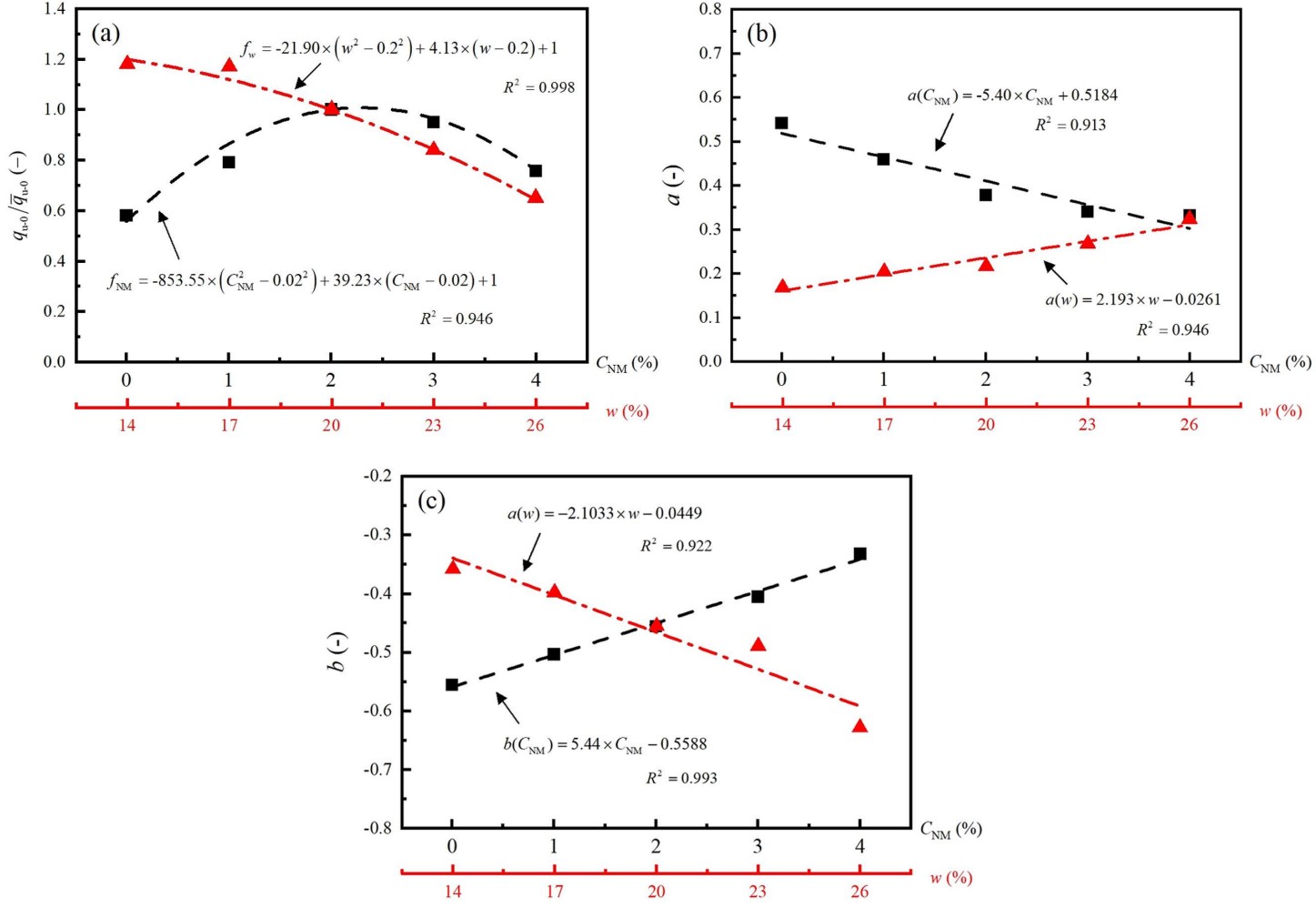

**Fig 9. Model parameters versus NM and water content: (a)** $q_{u\text{-}0}/\overline{q}_{u\text{-}0}$ **, (b)** $a$ **, (c)** $b$ **.**

Based on the formula, the effects of freeze-thaw cycles on the unconfined compressive strength of stabilized loess with different water contents can be estimated.

Fig 10 shows the tested and estimated UCS represented by bar figures and surface meshes, respectively. It is seen that the predictions by Eqs (7) and (8) agree well with tested results. The determination coefficients are acceptable values of 0.978 and 0.981, respectively. Therefore, it could be concluded that the proposed models are capable of predicting the UCS of NML under freezing-thawing circumstance.

### 3.5. NMR analysis

The results of NMR test for various conditions are displayed in Fig 11. In this figure, the red line represents the cutoff value of relaxation time. That is, the left/right side area under the $T_2$ spectra can be associated with the portion of bound water and free water, respectively. The cutoff $T_2$ of 1.65ms was adopted according to previous investigation on reconstituted loess [37,38]. Fig 11a illustrates the effects of FT cycle and water content on the $T_2$ spectra of 2% NM treated samples. With cyclic freezing-thawing, it is observed that $P_1$ peak of the $T_2$ spectra presented a downward shift, while $P_2$ peak shows an apparent upward shift. This phenomenon suggested a moisture migration towards less intensive interaction that is freezable under FT cycles. That is, FT cycles lead to higher proportion of free water inside soil, which can be attributed to the temperature gradient during freezing which results in an increase in water content in the frozen zone [39–42]. It is also noticed that NML samples with higher water content exhibited higher $P_1$ and $P_2$ peaks as expected. This FT-induced variation is more significant under higher water content. It in the effects of freeze-thaw was amplified with increasing water content, which is consistent with the results of UCT previously discussed.

Fig 11b presents the $T_2$ spectra for samples with varying FT cycles and NM contents. As expected, similar effects of FT cycles were observed. While, just opposed to the effects of water content, adding NM from 0% to 2% leads to an upward shift in $P_1$ peak and a downward shift in $P_2$ peak. The phenomenon indicates that the addition of NM increased the proportion of bound water and decreased the part of free water. This is attributed to the surface effect of the nanoparticles which

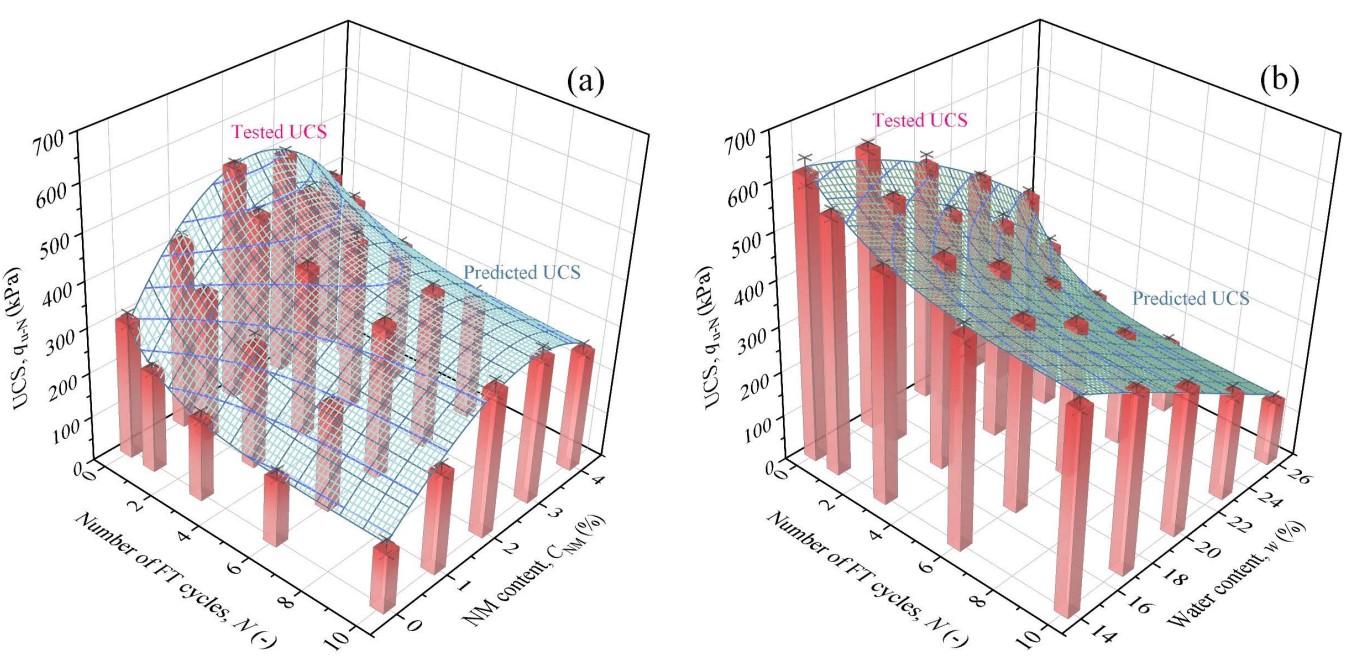

**Fig 10. Graphical representation of the proposed model: (a)** Eq (7)**, (b)** Eq (8)**.**

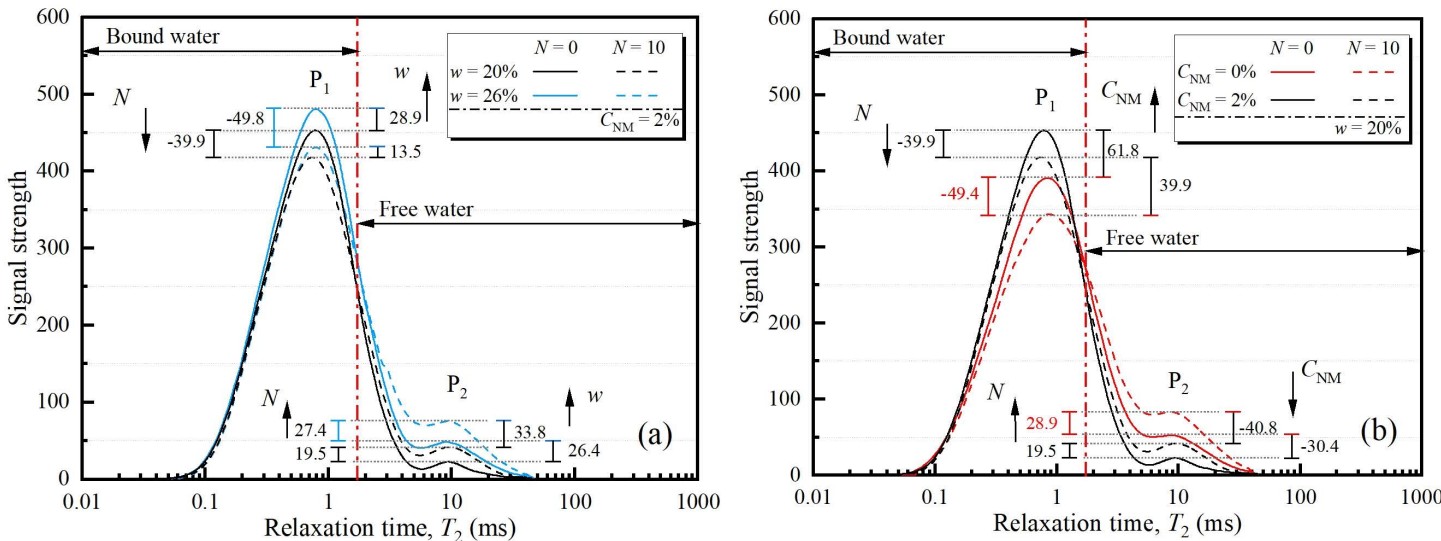

**Fig 11.** $T_2$ spectra for different FT cycles and (a) water contents; (b) NM contents.

results in higher proportion of bound water around NM nucleus during the hydration process. By comparing the results of different NM contents, it is found that the FT-induced decline in $P_1$ has narrowed from -49.4 to -39.9 as NM content from 0% to 2%, while the rise of $P_2$ has narrowed from -28.9 to -19.5. It implies that that the detrimental effects of freeze-thaw can be moderated by adding NM.

## 4. Conclusions

The present study investigated the issue of improving the mechanical performance of loess soil in cold regions with nano-MgO as a green and low carbon stabilizer. The strength and stiffness characteristics of the modified loess, as well as its freeze-thaw resistance, were investigated by conducting unconfined compressive test. Empirical models of mechanical parameters were proposed. Nuclear magnetic resonance test was invoked to explore the stabilizing mechanism. The main findings could be summarized as follows:

(1) Adding nano-MgO to loess soil enhanced the unconfined compressive strength of loess, while also reducing the failure strain under compression, and simultaneously increasing the deformation modulus. The peak strength of 2% nano-MgO stabilized loess soil exhibited the best performance of 71.9%, 98.1%, 111.6%, 128.1%, 143.5% strength gain for 0, 1, 3, 6, 10 FT cycles, respectively. Nano-MgO has higher stabilizing performance under FT circumstance.

(2) Ascending water content exerted negative effects on unconfined compressive strength and deformation modulus of loess, but increased the strain at failure. For a given nano-MgO content of 2%, the UCS values for water content of 14%, 17%, 20%, 23%, 26% respectively dropped 28.7%, 35.8%, 38.1%, 46.7%, 57.7% after 10 times of freezing and thawing, which signals the presence of mutually promotive interaction between water and FT cycles.

(3) The durability index, i.e., strength ratio of post- to pre-freeze-thaw samples, decreased exponentially with the number of freeze-thaw cycles. The durability index from the maximum FT cycles increased from 43.7% to 68.2% with NM content from 0% to 2%, and decreased from 55.1% to 32.7% as water content from 14% to 26%. That suggests the strength reduction resulting from freeze-thaw cycles can be alleviated by stabilizing loess with nano-MgO and preventing water intrusion in practice.

(4) Empirical models for UCS of NM treated loess, as well as its relationships with modulus and failure strain, were established by multiple regression and validated by literature data. These models can serve as a reference in the design of NM-treated soil in general and cold regions.

(5) NMRT test revealed ascending proportion of bound water with intensive interaction inside loess soil as the NM content increased, while the proportion of free water, that could bring strength loss during FT cycles, decreased. Therefore, the compressive behavior and freeze-thaw resistance of loess can be enhanced with suitable addition of NM.

Overall, the application of nano-MgO efficiently enhanced the strength, modulus and freeze-thaw resistance of loess soil, but also reduced the failure strain of loess, increasing the soil brittleness. Future studies should consider incorporate nano-MgO with different kinds of fibers to further enhance the soil strength and improve ductility.

## Supporting information

**S1 Data. UCS test data.**
(XLSX)

**S2 Data. NMR test data.**
(XLSX)

## Author contributions

**Conceptualization:** Shufeng Chen, Zhao Duan.

**Formal analysis:** Peng Hu, Shufeng Chen.

**Funding acquisition:** Shufeng Chen.

**Investigation:** Peng Hu.

**Methodology:** Peng Hu, Ye Hao.

**Software:** Xian Wang.

**Supervision:** Zhao Duan.

**Validation:** Ye Hao.

**Writing – original draft:** Peng Hu, Shufeng Chen.

**Writing – review & editing:** Zhao Duan, Nian-qin Wang.

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
