## [Decision Letter · Decision Letter 0]

14 Jan 2025

PONE-D-24-44491Effects of freeze-thaw cycles on mechanical performance of loess soil stabilized with nano magnesium oxidePLOS ONE

Dear Dr. Chen,

Thank you for submitting your manuscript to PLOS ONE. After careful consideration, we feel that it has merit but does not fully meet PLOS ONE’s publication criteria as it currently stands. Therefore, we invite you to submit a revised version of the manuscript that addresses the points raised during the review process.

We look forward to receiving your revised manuscript.

Kind regards,

Khalil Abdelrazek Khalil, Ph.D.

Academic Editor

PLOS ONE

3. We note that this submission includes NMR spectroscopy data. We would recommend that you include the following information in your methods section or as Supporting Information files:

1) The make/source of the NMR instrument used in your study, as well as the magnetic field strength. For each individual experiment, please also list: the nucleus being measured; the sample concentration; the solvent in which the sample is dissolved and if solvent signal suppression was used; the reference standard and the temperature.

2) A list of the chemical shifts for all compounds characterised by NMR spectroscopy, specifying, where relevant: the chemical shift (δ), the multiplicity and the coupling constants (in Hz), for the appropriate nuclei used for assignment.

3)The full integrated NMR spectrum, clearly labelled with the compound name and chemical structure.

We also strongly encourage authors to provide primary NMR data files, in particular for new compounds which have not been characterised in the existing literature. Authors should provide the acquisition data, FID files and processing parameters for each experiment, clearly labelled with the compound name and identifier, as well as a structure file for each provided dataset. See our list of recommended repositories here: https://journals.plos.org/plosone/s/recommended-repositories

“This research is supported by the National Natural Science Foundation of China (No. 12102367); the Special Fund for the Launch of Scientific Research in Xijing University (No. XJ23T01); the Horizontal Project of Shaanxi Coal Geology Group Co., Ltd. (No. SMDZ-2023CX-15).”

5. We are unable to open your Supporting Information file [Figures.zip]. Please kindly revise as necessary and re-upload.

Reviewers' comments:

Reviewer's Responses to Questions

**Comments to the Author**

1. Is the manuscript technically sound, and do the data support the conclusions?

Reviewer #1: Yes

Reviewer #2: Yes

2. Has the statistical analysis been performed appropriately and rigorously? 

Reviewer #1: Yes

Reviewer #2: Yes

3. Have the authors made all data underlying the findings in their manuscript fully available?

Reviewer #1: Yes

Reviewer #2: Yes

4. Is the manuscript presented in an intelligible fashion and written in standard English?

Reviewer #1: Yes

Reviewer #2: Yes

5. Review Comments to the Author

Reviewer #1: I have carefully reviewed the manuscript entitled “Effects of freeze-thaw cycles on mechanical performance of loess soil stabilized with nano magnesium oxide.” The paper is well-organized and provides valuable insights for researchers and engineers in the field of geotechnical engineering. However, there are significant concerns regarding the innovation, novelty, and interpretation of results that must be addressed before the manuscript can be considered for publication. Based on these concerns, my recommendation is "Major Revision." Below, I outline critical comments for the authors to consider:

1. Abstract:

The abstract offers a concise overview of the study but lacks clarity and specificity. The research objectives and key findings should be articulated more precisely. Additionally, numerical results, such as the percentage improvement in soil properties or freeze-thaw resistance, should be highlighted to enhance the impact.

2. Introduction:

o The introduction requires significant improvement. It should delve deeper into the advantages of nanomaterials compared to traditional stabilizers like cement and lime. I suggest citing the following reference for this discussion: doi.org/10.1016/j.jclepro.2022.135390.

o Introduce novel soil stabilization techniques, including Microbial Induced Calcite Precipitation (MICP). For this, refer to doi.org/10.1016/j.jrmge.2023.11.025.

o Freeze-thaw behavior is a critical aspect of geomaterial performance and has been extensively studied. For effects of freeze-thaw cycles on nano-stabilized soils, consider incorporating this reference: doi.org/10.12989/gae.2024.37.1.085.

o Conclude the introduction with a clear statement of the research's importance, novelty, and originality to strengthen the rationale for the study.

3. Materials and Methods:

o This section is well-structured and clear. However, in the "Experimental Program," presenting the key details in a tabular format would enhance readability and provide a quick reference for the experimental design.

o Provide more detailed explanations about the empirical expressions for Unconfined Compressive Strength (UCS) calculations to clarify their derivation and applicability.

4. Results and Discussion:

o The manuscript leans heavily toward reporting experimental data. To enhance the impact, include deeper analytical discussions and interpretations based on the results.

o Discuss the fitting results presented in Table 4 in greater detail. Explain how these results align with or differ from existing studies and provide possible reasons for the observed trends.

o After summarizing the key findings, compare them comprehensively with previous studies to highlight similarities, differences, and the unique contributions of your work.

5. Conclusion:

o The conclusion section should be expanded to include quantitative findings that summarize the improvements achieved in soil properties.

o Provide a more detailed and structured summary of the study’s implications for practical applications and future research directions.

6. Language and Readability:

o The manuscript would benefit significantly from a thorough language review by a native speaker or a professional editing service to improve its readability and polish.

By addressing the above comments, the manuscript can better position itself as a significant contribution to the field of soil stabilization using nanomaterials. I look forward to reviewing the revised submission.

Reviewer #2: In this study, the variation of mechanical properties of nano-MgO treated loess under freeze-thaw cycles was investigated through a series of unconfined compression tests. The topic of this research is interesting. However, the depth and novelty of this study needs to be further clarified. In addition, there are a number of grammatical and expression errors in the manuscript. The following issues are suggested to be revised before recommending publication.

1. The introduction does not highlight the innovation and necessity of this work. Many studies have investigated the effect of nano-MgO content and water content on the unconfined compressive strength of nano-MgO treated loess under freeze-thaw cycles.

2. In the unconfined compression tests, two samples were produced at each test condition. How to handle failure strength and strain data from two samples? The stress-strain curves of the samples were not present in the manuscript.

3. The description of the compaction test does not need to be included in Table 3.

4. In terms of mechanical properties and microstructure analysis, some relevant references may be referred, e.g., “Micro-scale investigations on the mechanical properties of expansive soil subjected to freeze-thaw cycles”, “Investigation of coupled thermo-hydro-mechanical processes on soil slopes in seasonally frozen regions”.

5. What is the use of prediction equations (7) and (8)?

6. In NMR analysis, what is non-freezable water and freezable water? Bound water and free water are used in many studies.

7. The freeze-thaw cycle can cause changes in the physical and mechanical properties of the rock and soil mass, the following literature might be referred.

(2024). Seismic Response and Dynamic Failure Mode of a Class of Bedding Rock Slope Subjected to Freeze–Thaw Cycles. Rock Mechanics and Rock Engineering, 1-19.

6. PLOS authors have the option to publish the peer review history of their article (what does this mean? ). If published, this will include your full peer review and any attached files.

**Do you want your identity to be public for this peer review?** For information about this choice, including consent withdrawal, please see our Privacy Policy .

Reviewer #1: **Yes: ** Meysam Bayat

Reviewer #2: No

---

## [Author Response · Author response to Decision Letter 1]

8 Feb 2025

We have respond to specific reviewer and editor comments as separate files.

---

## [Decision Letter · Decision Letter 1]

11 Feb 2025

Effects of freeze-thaw cycles on mechanical performance of loess soil stabilized with nano magnesium oxide

PONE-D-24-44491R1

Dear Dr. Chen,

We’re pleased to inform you that your manuscript has been judged scientifically suitable for publication and will be formally accepted for publication once it meets all outstanding technical requirements.

Kind regards,

Khalil Abdelrazek Khalil, Ph.D.

Academic Editor

PLOS ONE

Additional Editor Comments (optional):

Reviewers' comments:

Reviewer's Responses to Questions

**Comments to the Author**

1. If the authors have adequately addressed your comments raised in a previous round of review and you feel that this manuscript is now acceptable for publication, you may indicate that here to bypass the “Comments to the Author” section, enter your conflict of interest statement in the “Confidential to Editor” section, and submit your "Accept" recommendation.

Reviewer #1: All comments have been addressed

Reviewer #2: All comments have been addressed

2. Is the manuscript technically sound, and do the data support the conclusions?

Reviewer #1: Yes

Reviewer #2: Yes

3. Has the statistical analysis been performed appropriately and rigorously? 

Reviewer #1: Yes

Reviewer #2: Yes

4. Have the authors made all data underlying the findings in their manuscript fully available?

Reviewer #1: Yes

Reviewer #2: Yes

5. Is the manuscript presented in an intelligible fashion and written in standard English?

Reviewer #1: Yes

Reviewer #2: Yes

6. Review Comments to the Author

Reviewer #1: The new version of the paper is wholly modified compared to the original version, and the article is acceptable for publication.

Reviewer #2: I am quite satisfied with the authors response to my previous comments. The paper in its current form is now recommended acceptable for publication.

7. PLOS authors have the option to publish the peer review history of their article (what does this mean? ). If published, this will include your full peer review and any attached files.

**Do you want your identity to be public for this peer review?** For information about this choice, including consent withdrawal, please see our Privacy Policy .

Reviewer #1: **Yes: ** Meysam Bayat

Reviewer #2: No

---

## [Editor Report · Acceptance letter]

PONE-D-24-44491R1

PLOS ONE

Dear Dr. Chen,

I'm pleased to inform you that your manuscript has been deemed suitable for publication in PLOS ONE. Congratulations! Your manuscript is now being handed over to our production team.

Kind regards,

on behalf of

Dr. Khalil Abdelrazek Khalil

Academic Editor

PLOS ONE